# Timely initiation of antenatal care and its associated factors among pregnant women in sub-Saharan Africa: A multicountry analysis of Demographic and Health Surveys

**Adugnaw Zeleke Alem**[1]*, **Yigizie Yeshaw**[1,2], **Alemneh Mekuriaw Liyew**[1], **Getayeneh Antehunegn Tesema**[1], **Tesfa Sewunet Alamneh**[1], **Misganaw Gabrie Worku**[3], **Achamyeleh Birhanu Teshale**[1], **Zemenu Tadesse Tessema**[1]

1 Department of Epidemiology and Biostatistics, Institute of Public Health, College of Medicine and Health Sciences, University of Gondar, Gondar, Ethiopia, 2 Department of Physiology, School of Medicine, College of Medicine and Health Sciences, University of Gondar, Gondar, Ethiopia, 3 Department of Human Anatomy, College of Medicine and Health Science, School of Medicine, University of Gondar, Gondar, Ethiopia

* aduzeleke2201@gmail.com

## Abstract

### Background

Timely initiation of antenatal care (ANC) is an important component of ANC services that improve the health of the mother and the newborn. Mothers who begin attending ANC in a timely manner, can fully benefit from preventive and curative services. However, evidence in sub-Saharan Africa (sSA) indicated that the majority of pregnant mothers did not start their first visit timely. As our search concerned, there is no study that incorporates a large number of sub-Saharan Africa countries. Thus, the objective of this study was to assess the prevalence of timely initiation of ANC and its associated factors in 36 sSA countries.

### Methods

The Demographic and Health Survey (DHS) of 36 sSA countries were used for the analysis. The total weighted sample of 233,349 women aged 15–49 years who gave birth in the five years preceding the survey and who had ANC visit for their last child were included. A multilevel logistic regression model was used to examine the individual and community-level factors that influence the timely initiation of ANC. Results were presented using adjusted odds ratio (AOR) with 95% confidence interval (CI).

### Results

In this study, overall timely initiation of ANC visit was 38.0% (95% CI: 37.8–38.2), ranging from 14.5% in Mozambique to 68.6% in Liberia. In the final multilevel logistic regression model:- women with secondary education (AOR = 1.08; 95% CI: 1.06, 1.11), higher education (AOR = 1.43; 95% CI: 1.36, 1.51), women aged 25–34 years (AOR = 1.20; 95% CI: 1.17, 1.23), ≥35 years (AOR = 1.30; 95% CI: 1.26, 1.35), women from richest household (AOR = 1.19; 95% CI: 1.14, 1.22), women perceiving distance from the health facility as not

**Data Availability Statement:** The datasets we used for this study was existing public domain survey

data sets which accessed from http://www.
dhsprogram.com.

**Funding:** The author(s) received no specific
funding for this work.

**Competing interests:** The authors have declared
that no competing interests exist.

**Abbreviations:** ANC, Antenatal Care; AOR,
Adjusted Odds Ratio; CI, Confidence Interval; DHS,
Demographic and Health Survey; ICC, Intra-class
Correlation Coefficient; MMR, Maternal Mortality
Rate; MOR, Median Odds Ratio; PCV, Proportional
Change in Variance; sSA, sub-Saharan Africa.

a big problem (AOR = 1.05; 95%CI: 1.03, 1.07), women exposed to media (AOR = 1.29;
95%CI: 1.26, 1.32), women living in communities with medium percentage of literacy (AOR
= 1.51; 95%CI: 1.40, 1.63), and women living in communities with high percentage of liter-
acy (AOR = 1.56; 95%CI: 1.38, 1.76) were more likely to initiate ANC timely. However,
women who wanted their pregnancy later (AOR = 0.84; 95%CI: 0.82, 0.86), wanted no more
pregnancy (AOR = 0.80; 95%CI: 0.77, 0.83), and women residing in the rural area (AOR =
0.90; 95%CI: 0.87, 0.92) were less likely to initiate ANC timely.

## Conclusion

Even though the WHO recommends all women initiate ANC within 12 weeks of gestation,
sSA recorded a low overall prevalence of timely initiation of ANC. Maternal education, preg-
nancy intention, residence, age, wealth status, media exposure, distance from health facil-
ity, and community-level literacy were significantly associated with timely initiation of ANC.
Therefore, intervention efforts should focus on the identified factors in order to improve
timely initiation of ANC in sSA. This can be done through the providing information and edu-
cation to the community on the timing and importance of attending antenatal care and family
planning to prevent unwanted pregnancy, especially in rural settings.

## Background

Even though stillbirth rate (MMR) is reduced by 58.3% globally from 2000 to 2015, 98% of all
stillbirths occur in low and middle income (LMICs) countries. Of these, 77% occurs in sub
Saharan Africa (sSA) and south Asia [1]. Moreover, between 1990 and 2017, the global neona-
tal mortality rate (NMR) decreased from 36·6 deaths to 18.0 per 1000 live births. However, it is
difficult to achieve the Sustainable Development Goal (SDGs) NMR target of 12 deaths per
1000 live births or fewer by 2030 in most countries with current progress [2]. One of the targets
of the SDGs fixed by the United Nations is to end preventable maternal, under-five, and new-
borns deaths by the year 2030. These modifiable risk factors could be avoided through expand-
ing access to the mothers given during pregnancy like antenatal care (ANC) services [3]. It
reduces the risk of neonatal death by 39% in sSA [4]. Therefore, improving the utilization of
maternal health services is fundamental for SDG goal 3 achievements [5].

Many maternal and neonatal deaths can be prevented by expanding access to the care given
to the mother during pregnancy, during delivery, and after delivery [3, 6–8]. ANC is special
care given to pregnant women to timely identify and alleviate pregnancy-related complications
that can harm the mother and fetus. It is one of the essential strategies for reducing maternal
and child death directly or indirectly [6, 9–11]. Prevention and treatment of any complica-
tions/illness; emergency preparedness; birth planning; and health promotion like satisfying
any unmet nutritional, social, emotional, and physical needs of pregnant women, provision of
patient education are the main objectives of ANC [12–14]. Even though the percentage of
women attending ANC visit is increased even in low-income countries, in most sSA countries
maternal and neonatal mortality remain high [15–17]. This weak association between ANC
utilization and maternal and newborn survival has motivated a recent call to focus on quality
of ANC services rather than mere ANC attendance to ensure well being of fetus, mothers and
newborn [12]. Therefore, to make the ANC visits an effective preventive measure, ANC visit
should be initiated early and components of ANC should be provided.

In light of the above, 2016 WHO recommendations on ANC for a positive pregnancy experience modified the minimum number of ANC contacts from four to eight contacts, with the first contact should be done within the first 12 weeks of gestation [12, 18]. This new ANC model aims at increasing contacts from four to eight, highlights the critical need to further target women who initiate ANC late, to achieve the recommended eight contacts [12].

Timely initiation of ANC is one of the basic components of ANC services; that helps to early identification of pre-existing health conditions like HIV and other sexually transmitted diseases (STDs), malaria, and anemia and early detection of complications arising during pregnancy [14, 19–21]. Early screening and treatment of HIV and syphilis help to prevent maternal to fetus transmission. Untreated mothers have a 70–100% chance of transmitting the infection to their fetus, one-third pregnancies results in stillbirth and of the infants of mothers with untreated syphilis, 15% of had clinical evidence of congenital syphilis [22, 23]. Moreover, timely initiation of ANC is a good opportunity to discuss with pregnant mothers on birth preparedness and complication readiness plan and helps the mother to receive health promotion and disease prevention services such as immunization against tetanus, nutrition counseling, micro-nutrient supplements, prophylactic treatment of malaria and worms [9, 14, 21, 24]. The provision of micro-nutrient supplements especially iron and folic acid during early pregnancy is among the strongly recommended interventions in order to prevent anemia and congenital malformations [12]. Anemia is the lead cause of preterm births, and low birth weight, and maternal mortality [25, 26]. Existing evidence shows that the early initiation of ANC helps to ensure the well-being of the mother and fetus as well as their child [9, 24, 27]. Therefore, to fully benefit from ANC, it is important that women should start ANC timely.

Even though timely initiation of ANC is a key intervention for the reduction of maternal and child mortality, many pregnant women start ANC attendance late, particularly in sSA. Globally, the coverage of timely ANC initiation is around 43%, with a high discrepancy between developed and developing regions [28]. In developed regions, 85% of mothers start their ANC follow up in the first trimester compared to below 45% and less than 25% in the developing countries and sub-Sahara region respectively [29]. The timely initiation of ANC in sSA is low, as different Demographic Health Survey (DHS) reports the coverage of early timing of ANC visit ranges from 17.6 to 34% [30–33].

Previous studies that have investigated timely initiation and factors of ANC in sSA were mainly country-specific, with the focus on Ethiopia [6, 7, 11, 19, 24, 34–42], Nigeria [9, 43, 44], Uganda [14, 45], Tanzania [21, 46–49], Zambia [50–52], and Liberia [53, 54]. Our extensive search indicated that little evidence exists on the status of timely ANC initiation on the sSA scale. Besides, this study not only uses a rich source of data from sSA but also builds renewed evidence on the factors of timely initiation of ANC within the context of the agenda for SDG targets 3 to generate evidence-based decision making to improve timely initiation of ANC and maternal and child well being.

## Methods

### Study setting and design

The study used 36 sSA countries' Demographic and Health Survey (DHS) data which were obtained using a cross-sectional study design. The survey we used were conducted between 2006–07 and 2018 in sSA countries.

### Data source and sampling procedure

The data for this study were drawn from recent nationally representative DHS data conducted in 36 countries in sSA. The DHS surveys are routinely collected every five-year period across

low- and middle-income countries using structured methodologies and pretested validated quantitative tools. It follows the same standard procedure sampling, questionnaires, data collection, and coding which makes multi-country analysis possible.

In order to ensure national representativeness, the DHS survey employs a stratified two-stage sampling technique. In the first stage, clusters/enumeration areas (EAs) that cover the entire country were randomly selected from the sampling frame (i.e. are usually developed from the available latest national census). The second stage is the systematic sampling of households listed in each cluster or EA and interviews are conducted in selected households with target populations (women aged 15–49 and men aged 15–64). In this study, women aged 15–49 years who gave birth in the five years preceding the survey and who had ANC visit for their last child were included. The total sample size from the pooled data analyzed in this study was 233,349 and the sample size ranged from 1,316 in Sao Tome and Principe to 16,543 in Nigeria (**Table 1**).

## Variables of study

The outcome variable for this study was timely initiation of first ANC visit which was recorded as: within 12 weeks of gestation "timely" and after 12 weeks of gestation"delayed" [55].

Independent variables were extracted based on literature and the likelihood to influence the outcome of interest from the available DHS [6, 7, 9–11, 14, 19–21, 24, 34–42, 45, 46, 56, 57]. In this study, independent variables included in the analysis are broadly categorized as individual and community-level factors. The individual-level factors include maternal age (categorized as 15–24 years, 25–34 years, and ≥35 years), maternal education (no education, primary, secondary, and higher), marital status (categorized as ever married and never married), household wealth status was derived from a combination of all household variables describing housing and assets and computed using principal component analysis (poorest, poorer, middle, richer, and richest), media exposure (exposed to at least one of radio, magazine/newspaper or television were labeled as 'yes' and those who did not were labeled as 'no'), insurance coverage (yes/no), parity (categorized as primiparous, multiparous, and grand multiparous), ever had a pregnancy terminated (yes/no), pregnancy intention (wanted then, wanted later and wanted no more), perception of distance from the health facility (big problem/not a big problem) and employment status (not employed/employed).

Community-level factors were: place of residence (rural/urban), community-level literacy, community-level poverty, and community media exposure. The community-level variables such as community-level literacy, community-level poverty, and community media exposure were obtained by aggregating the individual-level variables into clusters by using the proportion. Community-level literacy is measured as the proportion of women who completed primary and above educational level in the primary sampling unit. It was categorized as low, medium and high if less than 25%, 25%-50% and more than 50% of study population of the cluster had at least eight years of education respectively. Community-level poverty was computed from the household wealth and defined as the proportion of women in the top 3 wealth quantiles (middle, richer and richest) in the clusters. It was categorized as low, medium and high if less than 25%, 25%-50% and more than 50% of study population of the cluster had at least middle quintile respectively. Community media exposure is the proportion of women who had exposure to at least one type of media; radio, newspaper, or television in the primary sampling unit. Similarly, community media exposure was categorized as low, medium, and high.

## Statistical analysis

All statistical analysis was carried out with STATA version 14. Since DHS surveys follows the same standard procedure sampling, questionnaires, data collection, and coding, datasets were

**Table 1. Countries, survey year, and samples of Demographic and Health Surveys included in the analysis for 36 sub-Saharan African countries.**

| Country | Survey year | Weighted sample size |
|---|---|---|
| Angola | 2015–16 | 6,919 |
| Burkina Faso | 2010 | 9,964 |
| Benin | 2017–18 | 7,965 |
| Burundi | 2016–17 | 8,867 |
| Central democratic Congo | 2013–14 | 9,918 |
| Congo | 211–12 | 5,474 |
| Cote d'vore | 2011–12 | 4,814 |
| Cameroon | 2018 | 5,758 |
| Ethiopia | 2016 | 4,741 |
| Gabon | 2012 | 3,518 |
| Ghana | 2014 | 4,034 |
| Gambia | 2013 | 5,252 |
| Guinea | 2018 | 4,689 |
| Kenya | 2014 | 13,839 |
| Comoros | 2012 | 1,879 |
| Liberia | 2013 | 4,632 |
| Lesotho | 2014 | 2,450 |
| Madagascar | 2008–09 | 7,794 |
| Mali | 2018 | 5,264 |
| Malawi | 2015–16 | 13,251 |
| Mozambique | 2011 | 7,112 |
| Nigeria | 2018 | 16,542 |
| Niger | 2012 | 6,817 |
| Namibia | 2013 | 3,693 |
| Rwanda | 2014–15 | 6,006 |
| Sera lone | 2013 | 8,372 |
| Senegal | 2010 | 7,238 |
| Sao tome and principe | 2008–09 | 1,316 |
| Eswatini | 2006–07 | 2,068 |
| Chad | 2014–15 | 7,050 |
| Togo | 2013 | 4,501 |
| Tanzania | 2015–16 | 6,930 |
| Uganda | 2016 | 9,947 |
| South Africa | 2016 | 2,845 |
| Zambia | 2018 | 7,233 |
| Zimbabwe | 2015 | 4,658 |

appended together to explore the timing of ANC and its associated factors among women in sSA. Both descriptive and analytic analysis were carried out after the weighting of data using sample weights to adjust disproportional sampling and non-response as well as to restore the representativeness of the sample so that the total sample looks like the country's actual population. Frequencies and percentages were used to describe the background characteristics of the study participants. Multilevel logistic regression was employed because our outcome variable (timing of the first ANC visit) was measured as a binary factor and since DHS data are hierarchical, i.e. individuals (level 1) were nested within communities (level 2). To cater for the unexplained variability at the community level, we used clusters as random effect. The log of the probability of the timing of ANC was modeled using a two-level model as follows:

$$\text{Log } [\Pi ij /1 - \Pi ij] = \beta 0 + \beta_1 Xij + B_2 Zij + \mu j + eij$$

Where

i and j are the individual (level 1) and community (level 2) units, respectively;

X and Z refer to level 1 and (level 2) variables, respectively;

$\pi ij$ is the probability of timely initiation of ANC

the β's are fixed coefficients;

β0 is the intercept-the effect on the probability of the timing of ANC in the absence of independent variables;

μj and eij are random effect (effect of the community on timing of ANC for the j[th] community) and random errors at the individual levels respectively.

In particular, three models were constructed [58]. We first constructed an empty model, which only includes outcome variable and cluster variable to test the random effect between-cluster variability. Then model containing only individual-level variables (model I) was fitted. Finally, in model II, we adjusted for both individual and community-level variables to estimate the association between timely initiation of ANC and the factors. The Intra-class Correlation Coefficient (ICC), the Median Odds Ratio (MOR), and the Proportional Change in Variance (PCV) were computed to assess the clustering effect/variability. ICC shows the variation in timely intiation of ANC for reproductive women due to community characteristics and it was calculated as follows:

ICC = VA/ (VA+3.29), where VA is the estimated variance of clusters in each model [59].

The MOR is defined as the median odds ratio between the area at highest risk and the area at the lowest risk when comparing two individuals from two different randomly chosen clusters. It was calculated using the formula:

$$\text{MOR} = \exp. [\sqrt{(2 \times VA)} \times 0.6745] \approx \exp(0.95\sqrt{VA})]$$

Where VA is the cluster level variance in each model [59, 60].

We used PCV to measure total variation attributed to an individual or/and community-level factors at each model. It was calculated as: PCV % = (VA−VB/VA)*100, where VA = variance of the empty model, and VB = variance of the model with more factors [59]. Moreover, deviance information criteria (DIC) was used to compare the candidate model, which was calculated as: deviance = -2log-likelihood ratio. It is always greater or equal than zero, being zero only if the fit is perfect. Therefore, model with the minimum value of deviance was selected for data analysis.

First, we fit unadjusted regression models for each explanatory variable to select variables for multivariable analysis, and variables with p-value ≤ 0.20 in the unadjusted regression analysis were included in multivariable analysis. Finally, results for the multivariable analysis have been presented as odds ratios (OR), with their corresponding 95% confidence intervals (CI), and p-value <0.05 were considered to be significant factors associated with the timely initiation of ANC.

## Ethics approval and consent to participate

Ethical approval for this study was not required since this study used existing public domain survey data sets, which are freely available online with all identifier information removed. But to access and use the data we sought permission and approval from Measure DHS through the online request.

## Results

### Background characteristics of respondents

A total of 233,349 reproductive-age women who gave birth in the five years preceding the survey and who attend ANC visits for their last pregnancy were included in this study. Most of

the participants were in the age range of 25–34 years (n = 107,454, 46.1%), not exposed media (n = 140,264, 60.1%) and rural residents (n = 151,955, 65.1%). Regarding pregnancy intention, nearly three quarters (n = 161,924, 71.6) of the respondents wanted their pregnancy later. Nearly, half of the study participants were from the community with high literacy (n = 115,402, 49.5%) and high media exposure (n = 113,363, 48.6%) (Table 2).

**Table 2. Background characteristics of the study participants.**

| Individual-level Variable | Frequency | Percentage | Community-level variable | Frequency | Percentage |
|---|---|---|---|---|---|
| Maternal age | | | Residence | | |
| 15–24 | 71,439 | 30.6 | Urban | 81,394 | 34.9 |
| 25–34 | 107,454 | 46.1 | Rural | 151,955 | 65.1 |
| ≥35 | 54,456 | 23.3 | Community-level literacy | | |
| Maternal education | | | Low | 59,508 | 25.5 |
| Not educated | 78,940 | 33.8 | Medium | 58,419 | 25.0 |
| Primary | 82,696 | 35.5 | High | 115,402 | 49.5 |
| Secondary | 62,378 | 26.7 | Community-level poverty | | |
| Higher | 9,335 | 4.0 | Low | 92,098 | 39.5 |
| Wealth status | | | Medium | 79,038 | 33.9 |
| Poorest | 45,322 | 19.4 | High | 62,193 | 26.6 |
| Poorer | 47,526 | 20.4 | Community media exposure | | |
| Middle | 47,297 | 20.2 | Low | 60,662 | 26.0 |
| Richer | 47,749 | 20.5 | Medium | 59,312 | 25.4 |
| Richest | 45,455 | 19.5 | High | 113,363 | 48.6 |
| Marital status | | | | | |
| Never married | 18,629 | 8.0 | | | |
| Ever married | 214,720 | 92.0 | | | |
| Employment status | | | | | |
| Not employed | 80,257 | 35.5 | | | |
| Employed | 145,616 | 64.5 | | | |
| Insurance coverage | | | | | |
| No | 196,754 | 93.7 | | | |
| Yes | 13,315 | 6.3 | | | |
| Ever had a pregnancy terminated | | | | | |
| No | 192,664 | 85.2 | | | |
| Yes | 33,443 | 14.8 | | | |
| Parity | | | | | |
| Primiparous | 51,434 | 22.1 | | | |
| Multiparous | 112,295 | 40.1 | | | |
| Grand multiparous | 69,620 | 29.8 | | | |
| Pregnancy intention | | | | | |
| Wanted then | 48,934 | 21.6 | | | |
| Wanted later | 161,924 | 71.6 | | | |
| No more | 15,224 | 6.8 | | | |
| Media exposure | | | | | |
| No | 140,264 | 60.1 | | | |
| Yes | 92,927 | 39.9 | | | |
| Distance from health facility | | | | | |
| Big problem | 83,519 | 38.6 | | | |
| Not big problem | 132,550 | 61.4 | | | |

## Prevalence of timely initiation of first ANC visit in sSA

The overall prevalence of timely initiation of ANC visit in 36 sSA countries was 38.0% (95% CI: 37.8–38.2). The prevalence of timely initiation of ANC visit was ranged from 14.5% in Mozambique to 68.6% in Liberia (**Fig 1**).

## Random effects and model comparison

The empty model indicates 4.2% of the total variation on timely initiation of ANC was at the cluster level and may be attributable to community-level factors (ICC = 0.042). In the final model (model II), the total variation on timely initiation of ANC at the cluster level was reduced to 2% (ICC = 0.002) and may be attributable to other unobserved community-level factors. Additionally, model II had the lowest MOR value (1.31) indicating the effects of community heterogeneity was low as compared with the empty model. In the final model (model II), as indicated by the PCV, 42.0% of the variation in timely initiation of ANC across communities was explained by both individual and community-level factors. Model II with the lowest (280,944) deviance was used to identify significantly associated factors with timely initiation of ANC among reproductive-age women in sSA (**Table 3**).

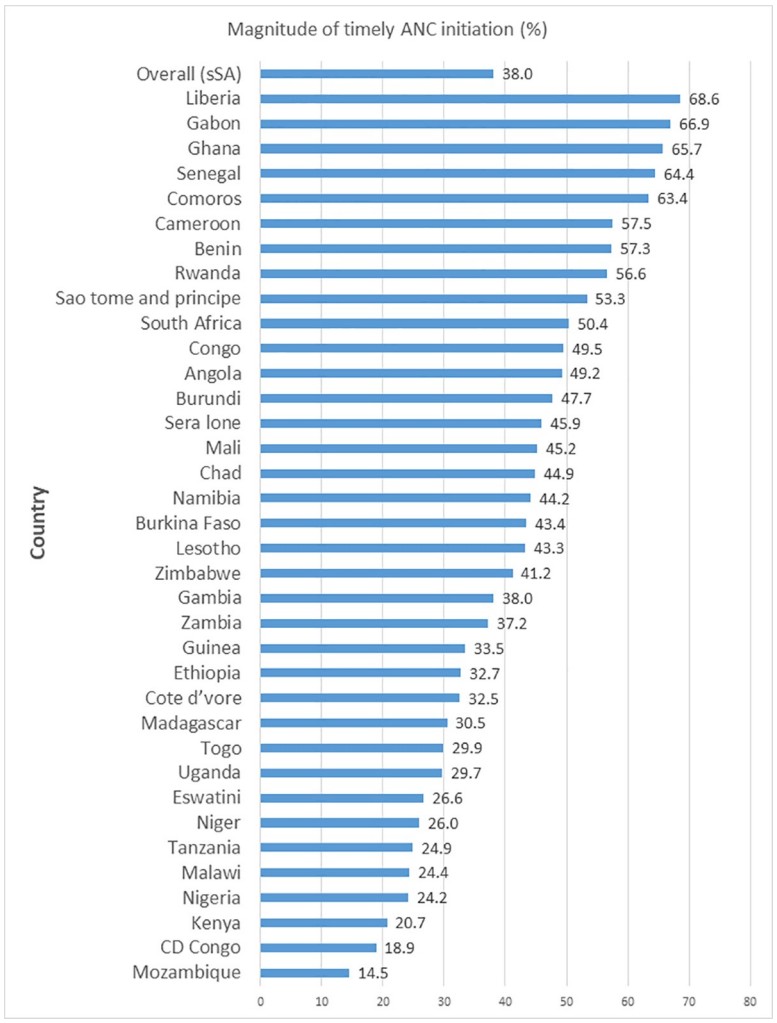

**Fig 1. Prevalence of timely initiation of ANC visit in sSA countries.**

**Table 3. Multilevel logistic regression analysis to assess factors associated with timely initiation of first ANC visit in reproductive-age women in sSA.**

| Variable | Empty model | Model I | Model II |
|---|---|---|---|
| | | AOR (95% CI) | AOR (95% CI) |
| Maternal age | | | |
| 15–24 | | 1 | 1 |
| 25–34 | | 1.20 (1.17,1.23) | 1.20 (1.17,1.22) |
| ≥35 | | 1.30 (1.26,1.34) | 1.30 (1.26,1.35) |
| Maternal education | | | |
| Not educated | | 1 | 1 |
| Primary | | 0.95 (0.93,0.97) | 0.96 (0.94,1.02) |
| Secondary | | 1.08 (1.05,1.11) | 1.08 (1.06,1.11) |
| Higher | | 1.41 (1.33,1.48) | 1.43 (1.36,1.51) |
| Wealth status | | | |
| Poorest | | 1 | 1 |
| Poorer | | 1.02 (0.99,1.05) | 1.01 (0.98,1.04) |
| Middle | | 1.01 (0.97,1.03) | 0.99 (0.98,1.02) |
| Richer | | 1.01 (0.98,1.04) | 0.98 (0.95,1.02) |
| Richest | | 1.25 (1.21,1.30) | 1.19 (1.14,1.22) |
| Employment status | | | |
| Not employed | | 1 | 1 |
| Employed | | 1.08 (1.06,1.10) | 1.03 (0.96,1.08) |
| Ever had a pregnancy terminated | | | |
| No | | 1 | 1 |
| Yes | | 1.15 (1.12,1.18) | 1.09 (0.98,1.11) |
| Parity | | | |
| Primiparous | | 1 | 1 |
| Multiparous | | 0.85 (0.83,0.87) | 0.95 (0.92,1.04) |
| Grand multiparous | | 0.66 (0.64,0.68) | 0.87 (0.84,1.03) |
| Pregnancy intention | | | |
| Wanted then | | 1 | 1 |
| Wanted later | | 0.81 (0.78,0.85) | 0.84 (0.82,0.86) |
| No more | | 0.74 (0.71,0.79) | 0.80 (0.77,0.83) |
| Media exposure | | | |
| No | | 1 | 1 |
| Yes | | 1.31 (1.29,1.34) | 1.29 (1.26,1.32) |
| Distance from HF | | | |
| Big problem | | 1 | 1 |
| Not a big problem | | 1.07 (1.04,1.08) | 1.05 (1.03,1.07) |
| Residence | | | |
| Urban | | | 1 |
| Rural | | | 0.90 (0.87,0.92) |
| Community media exposure | | | |
| Low | | | |
| Medium | | | 1.08 (0.97,1.05) |
| High | | | 1.01 (0.95,1.06) |
| Community-level litreacy | | | |
| Low | | | 1 |
| Medium | | | 1.51 (1.40,1.63) |
| High | | | 1.56 (1.38,1.76) |

*(Continued)*

**Table 3.** (Continued)

| Variable | Empty model | Model I | Model II |
|---|---|---|---|
| | | AOR (95% CI) | AOR (95% CI) |
| Community-level poverty | | | |
| Low | | | 1 |
| Medium | | | 1.02 (0.92,1.13) |
| High | | | 0.91 (0.80,1.03) |
| Community level variance | 0.143 | 0.124 | 0.083 |
| ICC | 0.042 | 0.036 | 0.02 |
| MOR | 1.43 | 1.40 | 1.31 |
| PCV | Reference | 13.3 | 42.0 |
| Deviance | 307,152 | 281,410 | 280,944 |

### Factors associated with timely initiation of ANC

As presented in **Table 3**, where both the individual and community-level factors were included simultaneously; age, maternal education, pregnancy intention, wealth status, distance from the health facility, and media exposure were individual-level factors significantly associated with timely initiation of ANC visit. Among community-level factors, residence and community-level literacy were significantly associated with timely initiation of ANC visit in reproductive-age women.

The odds of timely initiation of ANC was 1.20 (AOR = 1.20; 95% CI: 1.17, 1.23) and 1.30 (AOR = 1.30; 95% CI: 1.26,1.35) times higher among women aged 25–34 years and ≥35 years respectively as compared to women aged 15–24 years. The odd of timely initiation of ANC was 1.08 (AOR = 1.08; 95% CI: 1.06, 1.11) and 1.43 (AOR = 1.43; 95% CI: 1.36, 1.51) times higher in mothers who had secondary education and higher education respectively as compared to those mothers who had no formal education. Women in the richest wealth categories had 1.19 (AOR = 1.19; 95% CI: 1.14, 1.22) times higher odds of timely initiation of ANC as compared to the poorest women.

Higher odds of timely initiation of ANC occurred among women perceiving distance from the health facility as not a big problem as compared to women perceiving distance from the health facility as a big problem (AOR = 1.05; 95%CI: 1.03, 1.07). Again, higher odds occurred among women exposed to media as compared to women not exposed to media (AOR = 1.29; 95%CI: 1.26, 1.32).

The odds of timely initiation of ANC was 16% (AOR = 0.84; 95%CI: 0.82,0.86) and 20% (AOR = 0.80; 95%CI: 0.77, 0.83) lower among women who wanted their last pregnancy later and wanted no more pregnancy respectively as compared to women who wanted their last pregnancy then. A woman who was living in the rural area had 10% (AOR = 0.90; 95%CI: 0.87, 0.92) lower odds of timely initiation of ANC as compared with a woman who was living in urban areas. Moreover, the odds of timely initiation of ANC was 1.51 (AOR = 1.45; 95%CI: 1.40, 1.63) and 1.56 (AOR = 1.56; 95%CI: 1.38, 1.76) times higher among women living in communities with medium percentage of literacy and high percentage of literacy respectively as compared to women living in communites with low percentage of literacy.

## Discussion

Timely initiation of ANC is a key strategy for meeting new ANC model guidelines (the 2016 recommendation) for a positive pregnancy experience [7, 61]. This study provides information

on the timely initiation of ANC and its associated factors using data from DHS from 36 sub-Saharan African countries.

This study showed that 38.0% (95% CI: 37.8–38.2) of the reproductive women initiated their ANC within the recommended time with a wide range between countries ranged from 14.5% in Mozambique to 68.6% in Liberia. Also, a previous study in low- and middle-income countries (LMICs) reported that a wide range of timely initiation of ANC that ranged from 12.9% to 89.6% [62]. This variation among countries could be explained by the use of different cut-off points in defining early initiation of ANC (some countries defined early initiation of ANC based on the cut-off point of 12 weeks of gestation, whereas the other countries defined it based on 16 weeks), this results in a different perceived time of booking of ANC among women. However, this study used the same definition for all countries included in the analysis to defined timely initiation ANC based on the WHO definition. The WHO recommends women initiate first ANC visit within 12 weeks of gestation in order to achieve adequate ANC visits and identify and manage potential complications in early pregnancy [55]. However, this comprehensive analysis of 36 DHS data from the sSA countries suggested that a low proportion of women achieved this goal with variation among countries.

Consistent with previous studies conducted elsewhere [7, 9, 11, 20, 34, 39, 41–45], this study revealed that women with secondary and higher education were more likely to initiate ANC visit within the WHO recommended time (within 12 weeks of pregnancy) than women with no formal education. This is possibly because formal education increases women's understanding of multiple dimensions of health and health knowledge that leads women to seek greater use of acceptable maternal and child health services [63]. It is expected that educated women are more likely to understand the benefit of timely initiation of ANC visits and the negative effects of late ANC initiation. Moreover, educated women may have a high chance of exposure to information and have a greater decision making power on their own health as well as their children and demand higher quality service and pay more attention to their health in order to ensure better health for themselves and their child.

The results of this study show that higher odds of timely initiation of first ANC visit was observed among women in the richest wealth quantile. This finding is congruent with the study conducted in Ethiopia [11, 35, 39], Ghana [20], and Cameroon [56]. Despite ANC service is exempted service given for all pregnant women, it needs direct costs like transportation costs and indirect costs like household and work obligations in order to seek ANC [64]. Therefore, women with high household income are more likely to be able to afford the direct cost such as transportation costs and fulfill household and work obligations, this enables women to book for timely ANC. However, women with low household income need financial capacity to support their daily living and therefore they may be spent more time on economic activities to cater to their families rather than their health.

In agreement with existing literature [7, 9, 11, 19, 39–42, 45, 57, 65], this study indicated that women with unwanted pregnancy (pregnancy wanted later and no more) were less likely to start the first ANC visit within recommend time compared to women who wanted their pregnancy then. The possible reason might be women having wanted pregnancy has a chance of detecting the pregnancy earlier or women may give more cautious and excited to know their pregnancy status than those who had unwanted pregnancy. A study revealed that women who recognize their pregnancy earlier were more likely to early initiate the ANC services than those who recognize their pregnancy later [40]. Additionally, wanted pregnancy is more cared for by the pregnant women themselves and their spouses and a woman who prefers the pregnancy is willing to keep the health of the baby. Due to that, a woman with wanted pregnancy might seek appropriate care for their pregnancy and they are alerted about the advantage of attending timely.

In this study, we found that women younger than 24 years were less likely to timely start ANC visits compared to those who were older (25–34 years and ≥ 35 years). This finding supports the results of a study conducted in Ethiopia, Ghana, and Nigeria [20, 35, 39, 43]. This might be due to younger women who are unmarried are at risk of hesitating pregnancy disclosure to avoid potential social implications of the pregnancy. Because, in this study, 17.3% of young women (<25 years) were never married compared with 4.8% and 2.8% of women aged 25–34 years and ≥ 35 years. Furthermore, early pregnancy awareness/recognition is increased with maternal age [66]. This suggested that younger women are less likely to recognize their pregnancy early compared with older women. If pregnancy is recognized early, women might be prepared to initiate ANC timely as suggested in a study [40].

This study also found women who had media exposure/access was more likely to initiate ANC visit timely compared to those who had not. This finding is supported by the results of other studies conducted in Ethiopia [6] and Nigeria [44] which similarly showed that women with media exposure were more likely to start ANC timely. This might be because women who are exposed to media have better awareness and information on the existence of maternal health care services and the benefits of timely utilization of services. Evidence indicated that mass media is a critical source of health information globally, especially in LMICs. For example, in Nigeria, Tanzania, and Malawi mass media is one of the major sources of disseminating information to increase health knowledge and changing the health behaviors of women in order to improve maternal health [67–69].

The other factor that was associated with timely ANC visits was distance from the health facility. This study shows that women who consider the distance from the health facility as not a big problem was made their ANC visit timely. Similarly, a study conducted in Ethiopia [7, 19], Cameroon [56], and Uganda [45] reveal that the odds of timely initiation of ANC was higher among women who travel a long distance to reach a health facility. This might be due to financial constraints are in turn related to other barriers to seeking help, including transportation costs, the cost of obtaining care, or laboratory tests. This might be women who live distant to the maternity facility may impose an extra cost for transportation service as well as lack of availability of transportation and therefore they fail to attain the health facility for receiving ANC services timely [70].

Moreover, place of residence was found to have a significant association with timely initiation of first ANC visit. Those women from rural areas had lower odds of timely initiation ANC visit as compared with those who were from urban areas. This association was similar to studies done in Ethiopia, Uganda, Nigeria, and Malaysia [9, 34, 39, 42, 44, 45]. The early booking among women from urban areas is likely to be attributed to the adequate availability and accessibility of health facilities as well as health personnel and having a better chance of health information in urban areas than rural areas. The other reason might be in fact urban women have better educational status than rural women. In this study also, 41.4% of rural women were not attained formal education compared with 19.6% of urban women.

Apart from this, Community women's education was also found to be the factors positively associated with timely ANC commencement. The odds of timely initiation of ANC was higher among women from the community with medium education and high education as compared to women from the community with low education. This might be due to the high literacy level in the community that may cause high health knowledge in the community that increases adequate utilization of maternal health services like timely initiation of ANC visit.

## Strength and limitation of study

The main strength of the study was data used in this study were from nationally representative Demographic and Health Survey DHS from 36 sSA countries and therefore findings across the

sub-region could be generalized. The assessment of diverse factors such as individual and community-level factors that influences timely initiation of ANC using the multilevel analysis to accommodate the hierarchical nature of the data was another strength. Even though the important findings evolved in this study, the study had certain limitations that should be noted. The study was a cross-sectional study that did not show the temporal relationship between the outcome variable and independent variables. Moreover, recall bias might be a possible limitation because the DHS survey is relied on respondents' self-report based on their memories. Lastly, we acknowledge due to the secondary nature of data used important variables such as women's knowledge about the timing of ANC, women's perception on quality of ANC, and time of recognition of pregnancy were not included in this study.

## Conclusion

Even though the WHO recommends all women initiate ANC within 12 weeks of gestation, sSA recorded a low overall prevalence of timely initiation of ANC. Maternal education, pregnancy intention, residence, age, wealth status, media exposure, distance from the health facility, and community level literacy were significantly associated with timely initiation of ANC. Therefore, intervention efforts should focus on the identified factors in order to improve timely initiation of ANC in sSA. This can be done through the providing information and education to the community on the timing and importance of attending antenatal care and family planning to prevent unwanted pregnancy, especially in rural settings. Moreover, strategies should be designed to address the persistent health access inequity for younger and poorest women need to be prioritized in order for the countries to improve access to early initiation of ANC for a positive pregnancy experience and this intern improves maternal health and birth outcomes.

## Acknowledgments

The authors thank the Measure DHS program which granted us permission to use DHS data for this study.

## Author Contributions

**Conceptualization:** Adugnaw Zeleke Alem.

**Data curation:** Adugnaw Zeleke Alem, Yigizie Yeshaw, Alemneh Mekuriaw Liyew, Getayeneh Antehunegn Tesema, Tesfa Sewunet Alamneh, Misganaw Gabrie Worku, Achamyeleh Birhanu Teshale, Zemenu Tadesse Tessema.

**Formal analysis:** Adugnaw Zeleke Alem, Yigizie Yeshaw, Alemneh Mekuriaw Liyew, Getayeneh Antehunegn Tesema, Tesfa Sewunet Alamneh, Misganaw Gabrie Worku, Achamyeleh Birhanu Teshale, Zemenu Tadesse Tessema.

**Funding acquisition:** Alemneh Mekuriaw Liyew.

**Investigation:** Adugnaw Zeleke Alem, Yigizie Yeshaw, Alemneh Mekuriaw Liyew, Getayeneh Antehunegn Tesema, Tesfa Sewunet Alamneh, Misganaw Gabrie Worku, Achamyeleh Birhanu Teshale, Zemenu Tadesse Tessema.

**Methodology:** Adugnaw Zeleke Alem, Yigizie Yeshaw, Alemneh Mekuriaw Liyew, Getayeneh Antehunegn Tesema, Tesfa Sewunet Alamneh, Misganaw Gabrie Worku, Achamyeleh Birhanu Teshale, Zemenu Tadesse Tessema.

**Resources:** Adugnaw Zeleke Alem, Yigizie Yeshaw, Alemneh Mekuriaw Liyew, Getayeneh Antehunegn Tesema, Tesfa Sewunet Alamneh, Misganaw Gabrie Worku, Achamyeleh Birhanu Teshale, Zemenu Tadesse Tessema.

**Software:** Adugnaw Zeleke Alem, Yigizie Yeshaw, Alemneh Mekuriaw Liyew, Getayeneh Antehunegn Tesema, Tesfa Sewunet Alamneh, Misganaw Gabrie Worku, Achamyeleh Birhanu Teshale, Zemenu Tadesse Tessema.

**Supervision:** Yigizie Yeshaw, Alemneh Mekuriaw Liyew, Getayeneh Antehunegn Tesema, Tesfa Sewunet Alamneh, Misganaw Gabrie Worku, Achamyeleh Birhanu Teshale, Zemenu Tadesse Tessema.

**Validation:** Adugnaw Zeleke Alem, Yigizie Yeshaw, Alemneh Mekuriaw Liyew, Getayeneh Antehunegn Tesema, Tesfa Sewunet Alamneh, Misganaw Gabrie Worku, Achamyeleh Birhanu Teshale, Zemenu Tadesse Tessema.

**Visualization:** Adugnaw Zeleke Alem, Yigizie Yeshaw, Alemneh Mekuriaw Liyew, Getayeneh Antehunegn Tesema, Tesfa Sewunet Alamneh, Misganaw Gabrie Worku, Achamyeleh Birhanu Teshale, Zemenu Tadesse Tessema.

**Writing – original draft:** Adugnaw Zeleke Alem, Yigizie Yeshaw, Alemneh Mekuriaw Liyew, Getayeneh Antehunegn Tesema, Tesfa Sewunet Alamneh, Misganaw Gabrie Worku, Achamyeleh Birhanu Teshale, Zemenu Tadesse Tessema.

**Writing – review & editing:** Adugnaw Zeleke Alem, Yigizie Yeshaw, Alemneh Mekuriaw Liyew, Getayeneh Antehunegn Tesema, Tesfa Sewunet Alamneh, Misganaw Gabrie Worku, Achamyeleh Birhanu Teshale, Zemenu Tadesse Tessema.

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
