## [Decision Letter · Decision Letter 0]

14 Jun 2021

PONE-D-20-36739

Timely initiation of antenatal care and its associated factors among pregnant women in Sub-Saharan Africa: A multicounty analysis of Demographic and Health Surveys.

PLOS ONE

Dear Dr. Alem,

Thank you for submitting your manuscript to PLOS ONE. After careful consideration, we feel that it has merit but does not fully meet PLOS ONE’s publication criteria as it currently stands. Therefore, we invite you to submit a revised version of the manuscript that addresses the points raised during the review process. In particular, editing of the text and a number of point in the Mat and Meth section and results analysis section.

We look forward to receiving your revised manuscript.

Kind regards,

Isabelle Chemin, PhD

Academic Editor

PLOS ONE

Journal Requirements:

4. We noticed you have some minor occurrence of overlapping text with the following previous publications, which needs to be addressed:

- https://jepha.springeropen.com/articles/10.1186/s42506-020-00041-2 (Background, paragraph 3, sentence 2)

- http://jogh.org/documents/issue202001/jogh-10-010502.pdf (Background, paragraph 3, sentence 2)

- https://reproductive-health-journal.biomedcentral.com/articles/10.1186/s12978-017-0412-4 (Discussion, paragraph 8, sentence 4)

In your revision ensure you cite all your sources (including your own works), and quote or rephrase any duplicated text outside the methods section. Further consideration is dependent on these concerns being addressed.

Reviewers' comments:

Reviewer's Responses to Questions

**Comments to the Author**

1. Is the manuscript technically sound, and do the data support the conclusions?

Reviewer #1: Partly

2. Has the statistical analysis been performed appropriately and rigorously? 

Reviewer #1: Yes

3. Have the authors made all data underlying the findings in their manuscript fully available?

Reviewer #1: Yes

4. Is the manuscript presented in an intelligible fashion and written in standard English?

Reviewer #1: No

5. Review Comments to the Author

Reviewer #1: PONE-D-20-36739

Timely initiation of antenatal care and its associated factors among pregnant women in

Sub-Saharan Africa: A multicountry analysis of Demographic and Health Surveys.

This article describes timely initiation of antenatal care in sub-Saharan Africa and its associated factors. I think this is an important paper. However, I believe it would benefit from proof reading as I have noted several typos and grammatical errors. The methods seem strong but their descriptions needs to be improved.

Abstract

Background

1. “Mothers who attend ANC timely are fully benefited from its preventive and curative services.”

change to:

Mothers who begin attending ANC in a timely manner, can fully benefit from preventive and curative services.

Results

2. please use a consistent number of decimals throughout the paper. E.g.,: 37.99% (95% CI: 37.79-38.19), ranging from 14.50% in Mozambique to 68.60% in Liberia.

3. “women from the community with medium education (AOR = 1.45; 95%CI: 1.35, 1.51), and women from the community with high education (AOR = 1.50; 95%CI: 1.33, 1.68) “

Not clear what women from the community with medium education? Do you mean women living in communities in the 2nd tertile of education for example?

4. “However, women who wanted their pregnancy later (AOR = 0.84; 95%CI: 0.82 ,0.86), wanted no more (AOR = 0.80; 95%CI: 0.77, 0.83)”

I think there is a word missing after no more…

Conclusion:

5. SSA recorded a lower overall prevalence of timely initiation of ANC. Please change to:

SSA recorded a low overall prevalence of timely initiation of ANC.

6. “community women’s education”: please reword this is not clear.

7. “Therefore, more effort should be done to improve women’s education, wealth status of a household and family planning to prevent unwanted pregnancy and and then to improve the timely initiation of ANC.” This conclusion needs to be revised. Improving women’s wealth status is not an actionable strategy for public health practicioners. Please revise.

Background section

8. The background addresses maternal mortality at length. However, antenatal care does not have a strong relation to maternal deaths. Rather, it is quality intrapartum care that more strongly determines whether the mother survives. Quality ANC however, can have an important effect of fetal health and neonatal survival. Could the authors revise the background section to address the importance of ANC with more nuance, commenting on the importance of ANC to prevent miscarriages and stillbirths among others.

Please also remove the sentence: “ANC is the most important determinant of pregnancy outcomes”

9. The sentence: “85% of mothers start their ANC follow up earlier compared to below 45%...” is not clear. Do you mean that 85% start in the first trimester? Please revise.

Methods

10. Can you please describe how you selected these 36 surveys? Did you have a cut off date? Did you include all SSA countries?

11. Please change “ever had of a terminated pregnancy” to ever had a pregnancy terminated

12. Please change working status to employment status

13. perception of distance from the health facility (big problem/not a big problem) is not a community level factor. It is an individual level factor that is influenced by the woman’s ability to travel.

14. Please explain how you calculated community-level wealth. The following sentence is confusing: the proportion of women in the poorest and poorer quantiles in the community. What are “poorer” quintiles? 1 to 4?

Statistical analysis

15. The multi-level model is not clear. The authors should describe whether they used a random intercept or random slope model. and what level of clustering was used? The PSU, region or the country?

16. The first model (empty model) should contain the random effect. Is that what the authors did? Please revise the description.

17. Can you explain what deviance is? And why is model III in brackets after this sentence?

“To select the best-fitted model deviance was used and the

model with the lowest deviance was selected (model III).”

18. Please explain how you obtained the ICC (also known as variance partition coefficient in a logistic model)

19. Please describe what you mean by Median Odds Ratio (and include a reference for readers not familiar with the term).

20. I don’t see any benefit to running model 2 separately (community level variables only). I would recommend model 1: empty model, model 2: model with individual variables only and finally full model 3 with individual and community variables. The authors can then comment on the change in community-level variable as variables are added in. please cite: Intermediate and advanced topics in multilevel logistic regression analysis Peter C. Austin and Juan Merlo

Results

21. How did you pool results across countries? DHS sampling weights are not meant for multi-country comparisons. Each country should either be weighted equally or should be re-weighted based on population size for example. Another option would be to list result as: exposure to media ranged from X% in Botswana to X% in Rwanda.

22. Same comment for the regression models. Did you include sampling weights? Please also explain whether you pooled all the data together or ran models separately in every country…

23. This doesn’t seem correct: Most participants had attained higher education (n = 9,335, 40.0%). According to a quick internet search: only 6 percent of people in sub-Saharan Africa are enrolled in higher education institutions compared to the global average of 26 percent. Source: http://www.aaionline.org/wp-content/uploads/2015/09/AAI-SOE-report-2015-final.pdf

24. Replace three-fourth to three quarters

25. Table 2: According to table 2 it is 4% that have achieved higher educaiton. Not 40%. Please review table 2 and make sure all calculations are correct.

26. Community women’s education: in the table is described as low, medium, high. However, this is simply the % of women in the community who have achieved primary school. This is misleading, as primary education is not a “high” education achievement. Therefore I would recommend replacing with first, second and third tertile. Or by the actual range of % of women who achieved primary school e.g., <15%, 15%-40%, >40%.

27. Prevalence of timely initiation of first ANC visit in SSA: replace counties to countries.

28. Figure 1: please rank the countries by timely ANC initiation.

29. In table 3, please insert” reference” for PCV under the 1st model.

Discussion

30. The conclusions and policy implications need substantial improvement. Improving women’s wealth status is not an actionable strategy for public health practitioners. Please remove this recommendation. The authors should discuss what can be done to promote earlier access to antenatal care in SSA and the kinds of strategies to target poorer/less educated women and their communities, more at risk to starting ANC late. There must be literature on this. And others must have tested strategies to achieve this.

31. In the background and discussion sections, please also expand on the benefits for mothers, but particularly newborns, of starting ANC in the first trimester of gestation. For example, detecting and treating different infections (e.g., malaria, syphilis, HIV etc.) early in the pregnancy can have substantial effects on improving fetal and newborn outcomes. Providing nutritional supplements and vitamins can also improve fetal and newborn outcomes. E.g., folic acid in the first trimester is crucial.

32. Please add a paragraph on the need for ANC to be of good quality in order to have any effect. Poor quality ANC, even if started early, is unlikely to improve health outcomes. There are many references on quality of antenatal care and inequities in ANC access and quality.

Other references possibly of interest: https://www.ncbi.nlm.nih.gov/pmc/articles/PMC7457794/#:~:text=Parity%2C%20number%20of%20alive%20children,pregnancy%20approval%20by%20a%20spouse.

https://bmcpregnancychildbirth.biomedcentral.com/articles/10.1186/1471-2393-14-287

https://www.hindawi.com/journals/aph/2017/1624245/

https://journals.plos.org/plosone/article?id=10.1371/journal.pone.0246230

6. PLOS authors have the option to publish the peer review history of their article (what does this mean?). If published, this will include your full peer review and any attached files.

Reviewer #1: No

---

## [Author Response · Author response to Decision Letter 0]

24 Jul 2021

Rebuttal letter Date 7/ 24/2021

PONE-D-20-36739

Title: Timely initiation of antenatal care and its associated factors among pregnant women in

Sub-Saharan Africa: A multicountry analysis of Demographic and Health Surveys.

Editor comments

Version 1

Adugnaw Zeleke Alem

 Yigizie Yeshaw

 Alemneh Mekuriaw Liyew

 Getayeneh Antehunegn Tesema

 Tesfa Sewunet Alamneh

 Misganaw Gabrie Worku

 Achamyeleh Birhanu Teshale

 Zemenu Tadesse Tessema

Dear Editor and reviewer,

We would like to thank for your consideration and suggestion to improve our paper to make it more informative study. We tried to address all suggestions and clarification questions of editor and reviewer on the manuscript. Our point-by-point responses for each comment and questions are described in detail on the following pages. Further, the details of changes were shown by track changes in the supplementary document attached. 

Editor comments

1. Please ensure that your manuscript meets PLOS ONE's style requirements, including those for file naming. The PLOS ONE style templates can be found at https://journals.plos.org/plosone/s/file?id=wjVg/PLOSOne_formatting_sample_main_body.pdf andhttps://journals.plos.org/plosone/s/file?id=ba62/PLOSOne_formatting_sample_title_authors_affiliations.pd

Authors response: Thank you very much for your valuable references, we have prepared our manuscript based on PLOS ONE style.

Authors response: We have made an extensive edition on it with the help of English language expert (see track change).

Authors response: Thank you, I have linked My ORCID to editorial manager account

4. We noticed you have some minor occurrence of overlapping text with the following previous publications, which needs to be addressed:

- https://jepha.springeropen.com/articles/10.1186/s42506-020-00041-2 (Background, paragraph 3, sentence 2)

- http://jogh.org/documents/issue202001/jogh-10-010502.pdf (Background, paragraph 3, sentence 2)

- https://reproductive-health-journal.biomedcentral.com/articles/10.1186/s12978-017-0412-4 (Discussion, paragraph 8, sentence 4)

In your revision ensure you cite all your sources (including your own works), and quote or rephrase any duplicated text outside the methods section. Further consideration is dependent on these concerns being addressed.

Authors response: This has been cited and rephrased.

Reviewer comments 

Abstract

Background

1. “Mothers who attend ANC timely are fully benefited from its preventive and curative services.” change to: Mothers who begin attending ANC in a timely manner, can fully benefit from preventive and curative services.

Authors response: This has been changed. 

Results

2. please use a consistent number of decimals throughout the paper. E.g.,: 37.99% (95% CI: 37.79-38.19), ranging from 14.50% in Mozambique to 68.60% in Liberia.

Authors response: Thank you reviewer, even though there is no hard rule for decimals, using one or two decimal place suitable within text because readers can more easily understand numbers with fewer decimal places reported. The Cochrane Style and other guide lines recommends two decimal places for reporting odds ratios and risk ratios (inferential statistics) and one decimal place for proportion. Therefore, we have used two decimal places for odds ratio, otherwise one decimal place throughout the paper in the revised manuscript.

3. “women from the community with medium education (AOR = 1.45; 95%CI: 1.35, 1.51), and women from the community with high education (AOR = 1.50; 95%CI: 1.33, 1.68) “

 Not clear what women from the community with medium education? Do you mean women living in communities in the 2nd tertile of education for example?

Authors response: This has been clarified.

4. “However, women who wanted their pregnancy later (AOR = 0.84; 95%CI: 0.82 ,0.86), wanted no more (AOR = 0.80; 95%CI: 0.77, 0.83)” I think there is a word missing after no more…

Authors response: “Pregnancy” has been added after no more.

Conclusion: 

5. SSA recorded a lower overall prevalence of timely initiation of ANC. Please change to: 

SSA recorded a low overall prevalence of timely initiation of ANC.

Authors response: This has been changed.

6. “community women’s education”: please reword this is not clear.

Authors response: “community women’s education” has been replaced by “community-level literacy”.

7. “Therefore, more effort should be done to improve women’s education, wealth status of a household and family planning to prevent unwanted pregnancy and then to improve the timely initiation of ANC.” This conclusion needs to be revised. Improving women’s wealth status is not an actionable strategy for public health practitioners. Please revise.

Authors response: This has been revised (abstract section, page 3, line 52-55)

Background section

8. The background addresses maternal mortality at length. However, antenatal care does not have a strong relation to maternal deaths. Rather, it is quality intrapartum care that more strongly determines whether the mother survives. Quality ANC however, can have an important effect of fetal health and neonatal survival. Could the authors revise the background section to address the importance of ANC with more nuance, commenting on the importance of ANC to prevent miscarriages and stillbirths among others.

 Please also remove the sentence: “ANC is the most important determinant of pregnancy outcomes”

Authors response: This has been considered in the revised paper The sentence: “85% of mothers start their ANC follow up earlier compared to below 45%...” is not clear. Do you mean that 85% start in the first trimester? Please revise.

Authors response: This has been revised (background section, page 6, line 109-110)

Methods

9. Can you please describe how you selected these 36 surveys? Did you have a cutoff date? Did you include all SSA countries?

Authors response: We used all datasets available for sSA countries collected in year later 2002 considering introduction of 2002 of the WHO ANC model, known as focused ANC (FANC) or basic ANC, which was a goal orientated approach to delivering evidence-based interventions carried out at four critical times during pregnancy

10. Please change “ever had of a terminated pregnancy” to ever had a pregnancy terminated

Authors response: “ever had of a terminated pregnancy” has been changed to “ever had a pregnancy terminated”.

11. Please change working status to employment status

Authors response: “working status” has been changed to “employment status”.

12. perception of distance from the health facility (big problem/not a big problem) is not a community level factor. It is an individual level factor that is influenced by the woman’s ability to travel. 

Authors response: We have considered distance from health facility as individual level variable in revised manuscript.

13. Please explain how you calculated community-level wealth. The following sentence is confusing: the proportion of women in the poorest and poorer quantiles in the community. What are “poorer” quintiles? 1 to 4?

Authors response: This has been clarified (methods section, page 10, line 169-172).

Statistical analysis

14. The multi-level model is not clear. The authors should describe whether they used a random intercept or random slope model. and what level of clustering was used? The PSU, region or the country?

Authors response: This has been clarified (methods section, page 10 & 11, line 184-192).

15. The first model (empty model) should contain the random effect. Is that what the authors did? Please revise the description.

Authors response: This has been revised (methods section, page 11, line 199-200).

16. Can you explain what deviance is? And why is model III in brackets after this sentence?

“To select the best-fitted model deviance was used and the model with the lowest deviance was selected (model III).” 

Authors response: This has been elaborated (methods section, page 12, line 216-219).

17. Please explain how you obtained the ICC (also known as variance partition coefficient in a logistic model)

Authors response: This has been revised (methods section, page 11, line 205-208).

18. Please describe what you mean by Median Odds Ratio (and include a reference for readers not familiar with the term).

Authors response: This has been described and cited in revised manuscript (methods section, page 12, line 209-213).

19. I don’t see any benefit to running model 2 separately (community level variables only). I would recommend model 1: empty model, model 2: model with individual variables only and finally full model 3 with individual and community variables. The authors can then comment on the change in community-level variable as variables are added in. please cite: Intermediate and advanced topics in multilevel logistic regression analysis Peter C. Austin and Juan Merlo

Authors response: Model which includes only community level variable has been removed and Intermediate and advanced topics in multilevel logistic regression analysis Peter C. Austin and Juan Merlo has been cited in the revised manuscript accordingly (methods section, page 11, line 199-203).

Results

20. How did you pool results across countries? DHS sampling weights are not meant for multi-country comparisons. Each country should either be weighted equally or should be re-weighted based on population size for example. Another option would be to list result as: exposure to media ranged from X% in Botswana to X% in Rwanda. 

Authors response: Due to the non-proportional allocation of the sample to the different regions of countries and the possible differences in response rates, sampling weights are required for any analysis using the DHS data to ensure the representativeness of the survey results at the national as well as the regional level of each country. Since DHS surveys follows the same standard procedure sampling, questionnaires, data collection, and coding, all datasets were appended together for analysis. Therefore, we executed the svy command in the pooled datasets, including unique codes for each country's primary sampling unit and strata. 

21. Same comment for the regression models. Did you include sampling weights? Please also explain whether you pooled all the data together or ran models separately in every country… 

Authors response: Also, Sampling weight was performed for regression models and we have performed regression models for appended data (methods section, page 10, line 177-183).

22. This doesn’t seem correct: Most participants had attained higher education (n = 9,335, 40.0%). According to a quick internet search: only 6 percent of people in sub-Saharan Africa are enrolled in higher education institutions compared to the global average of 26 percent. Source: http://www.aaionline.org/wp-content/uploads/2015/09/AAI-SOE-report-2015-final.pdf

Authors response: Thank you very much for this concern. We agree that it is not 40.0%. This result was due to calculation error. Correction has been taken in the revised manuscript. Based on our result only 4% (n = 9,335) of participants had attained higher (see table 2).

23. Replace three-fourth to three quarters

Authors response: This has been replaced.

24. Table 2: According to table 2 it is 4% that have achieved higher education. Not 40%. Please review table 2 and make sure all calculations are correct. 

Authors response: All calculation has been reviewed.

25. Community women’s education: in the table is described as low, medium, high. However, this is simply the % of women in the community who have achieved primary school. This is misleading, as primary education is not a “high” education achievement. Therefore, I would recommend replacing with first, second and third tertile. Or by the actual range of % of women who achieved primary school e.g., <15%, 15%-40%, >40%.

Authors response: This has been elaborated (methods section, page 9 & 10, line 166-169).

26. Prevalence of timely initiation of first ANC visit in SSA: replace counties to countries.

Authors response: This has been replaced.

27. Figure 1: please rank the countries by timely ANC initiation.

Authors response: This has been considered

28. In table 3, please insert” reference” for PCV under the 1st model.

Authors response: Reference has been added under the empty model.

Discussion

29. The conclusions and policy implications need substantial improvement. Improving women’s wealth status is not an actionable strategy for public health practitioners. Please remove this recommendation. The authors should discuss what can be done to promote earlier access to antenatal care in SSA and the kinds of strategies to target poorer/less educated women and their communities, more at risk to starting ANC late. There must be literature on this. And others must have tested strategies to achieve this.

Authors response: This has been modified (conclusion section, page 24, line 401-408).

30. In the background and discussion sections, please also expand on the benefits for mothers, but particularly newborns, of starting ANC in the first trimester of gestation. For example, detecting and treating different infections (e.g., malaria, syphilis, HIV etc.) early in the pregnancy can have substantial effects on improving fetal and newborn outcomes. Providing nutritional supplements and vitamins can also improve fetal and newborn outcomes. E.g., folic acid in the first trimester is crucial.

Authors response: This has been elaborated (background section, page 5, line 89-105).

31. Please add a paragraph on the need for ANC to be of good quality in order to have any effect. Poor quality ANC, even if started early, is unlikely to improve health outcomes. There are many references on quality of antenatal care and inequities in ANC access and quality. Other references possibly of interest: https://www.ncbi.nlm.nih.gov/pmc/articles/PMC7457794/#:~:text=Parity%2C%20number%20of%20alive%20children,pregnancy%20approval%20by%20a%20spouse.

https://bmcpregnancychildbirth.biomedcentral.com/articles/10.1186/1471-2393-14-287

https://www.hindawi.com/journals/aph/2017/1624245/

https://journals.plos.org/plosone/article?id=10.1371/journal.pone.0246230

Authors response: Thank you very much for your sharing valuable references. The importance of quality/contents of ANC has been added (background section, page 4 & 5, line 77-83 & 89-105).

---

## [Editor Report · Decision Letter 1]

24 Dec 2021

Timely initiation of antenatal care and its associated factors among pregnant women in sub-Saharan Africa: A multicountry analysis of Demographic and Health Surveys.

PONE-D-20-36739R1

Dear Dr. Alem,

We’re pleased to inform you that your manuscript has been judged scientifically suitable for publication and will be formally accepted for publication once it meets all outstanding technical requirements.

Kind regards,

Isabelle Chemin, PhD

Academic Editor

PLOS ONE

Additional Editor Comments (optional):

The manuscript was significantly improved after the first round of review
---

## [Editor Report · Acceptance letter]

31 Dec 2021

PONE-D-20-36739R1 

Timely initiation of antenatal care and its associated factors among pregnant women in sub-Saharan Africa: A multicountry analysis of Demographic and Health Surveys. 

Dear Dr. Alem:

I'm pleased to inform you that your manuscript has been deemed suitable for publication in PLOS ONE. Congratulations! Your manuscript is now with our production department. 

Kind regards, 

on behalf of

Mrs Isabelle Chemin 

Academic Editor

PLOS ONE